# Development and Validation of a Highly Sensitive Multiplex Immunoassay for SARS-CoV-2 Humoral Response Monitorization: A Study of the Antibody Response in COVID-19 Patients with Different Clinical Profiles during the First and Second Waves in Cadiz, Spain

**DOI:** 10.3390/microorganisms11122997

**Published:** 2023-12-16

**Authors:** Lucia Olvera-Collantes, Noelia Moares, Ricardo Fernandez-Cisnal, Juan P. Muñoz-Miranda, Pablo Gonzalez-Garcia, Antonio Gabucio, Carolina Freyre-Carrillo, Juan de Dios Jordan-Chaves, Teresa Trujillo-Soto, Maria P. Rodriguez-Martinez, Maria I. Martin-Rubio, Eva Escuer, Manuel Rodriguez-Iglesias, Cecilia Fernandez-Ponce, Francisco Garcia-Cozar

**Affiliations:** 1Department of Biomedicine, Biotechnology and Public Health, School of Medicine, University of Cadiz, 11003 Cadiz, Spainantonio.gabucio@uca.es (A.G.); manuel.rodrigueziglesias@uca.es (M.R.-I.); 2Institute of Biomedical Research Cadiz (INIBICA), 11009 Cadiz, Spain; 3Microbiology Service, Puerto Real University Hospital, 11510 Puerto Real, Spain; 4Microbiology Service, Puerta del Mar University Hospital, 11009 Cadiz, Spain; 5La Milagrosa Health Centre, 11406 Jerez de la Frontera, Spain; 6Jerez Costa Noroeste Health District, 11403 Jerez de la Frontera, Spain; 7Jerez University Hospital, 11407 Jerez de la Frontera, Spain; eva.escuer.sspa@juntadeandalucia.es

**Keywords:** SARS-CoV-2, COVID-19, antibody response, IgG antibodies, multiplex immunoassay, monitoring tool

## Abstract

There is still a long way ahead regarding the COVID-19 pandemic, since emerging waves remain a daunting challenge to the healthcare system. For this reason, the development of new preventive tools and therapeutic strategies to deal with the disease have been necessary, among which serological assays have played a key role in the control of COVID-19 outbreaks and vaccine development. Here, we have developed and evaluated an immunoassay capable of simultaneously detecting multiple IgG antibodies against different SARS-CoV-2 antigens through the use of Bio-Plex^TM^ technology. Additionally, we have analyzed the antibody response in COVID-19 patients with different clinical profiles in Cadiz, Spain. The multiplex immunoassay presented is a high-throughput and robust immune response monitoring tool capable of concurrently detecting anti-S1, anti-NC and anti-RBD IgG antibodies in serum with a very high sensitivity (94.34–97.96%) and specificity (91.84–100%). Therefore, the immunoassay proposed herein may be a useful monitoring tool for individual humoral immunity against SARS-CoV-2, as well as for epidemiological surveillance. In addition, we show the values of antibodies against multiple SARS-CoV-2 antigens and their correlation with the different clinical profiles of unvaccinated COVID-19 patients in Cadiz, Spain, during the first and second waves of the pandemic.

## 1. Introduction

During the coronavirus disease 2019 (COVID-19) pandemic, caused by severe acute respiratory syndrome coronavirus 2 (SARS-CoV-2), the need for the development of rapid, specific and sensitive detection methods was apparent, being essential for the establishment of an effective prevention and treatment protocol, as well as for the epidemiological monitoring of the disease [1,2,3,4,5,6,7].

SARS-CoV-2 is a virus with a single-stranded, positive-sense RNA genome (+ssRNA) [8]. It consists primarily of four structural proteins: the spike (S) glycoprotein, envelope (E) protein, membrane (M) glycoprotein and nucleocapsid (NC) phosphoprotein [9]. The SARS-CoV-2 spike glycoprotein presents key mutations not found in other coronaviruses, which plays a crucial role in its high infectivity. This protein contains a receptor-binding domain (RBD) in the S1 subunit, responsible for binding to the ACE2 receptor on the surface of host cells, and subsequently allowing the fusion of the virus and cell membranes, which leads to cell infection [10,11,12].

Most serological tests used for monitoring immune responses against SARS-CoV-2 use one single antigen, such as the NC protein, RBD or S glycoprotein, to be recognized by antibodies present in patient sera [13,14,15]. As any test can lead to false positives due to sequence similarity between SARS-CoV-2 and other coronaviruses, like SARS-CoV or seasonal human coronavirus [16,17,18], relying on a multiantigen test may constitute a better option.

This study aims to develop and evaluate a serological multiplex assay capable of simultaneously detecting anti-S1, anti-RBD and anti-NC immunoglobulin G (IgG) antibodies in serum based on the Bio-Plex^TM^ multiplex assay platform, which is a high-throughput tool able to detect several analytes at the same time [19,20]. The immunoassay designed herein makes use of fluorescence-coded magnetic beads, functionalized with either S1, NC or RBD, placed in contact with patient sera to capture the anti-SARS-CoV-2 IgG antibodies that will be detected by means of a labeled anti-IgG antibody.

With the aim of evaluating the sensitivity, specificity and overall quality of the proposed antibody-based immune response monitoring tool for COVID-19, a receiver operating characteristic (ROC) curve analysis was carried out. Moreover, we have studied the relationship between antibody levels against multiple SARS-CoV-2 antigens and the different clinical profiles of unvaccinated COVID-19 patients in Cadiz, Spain, during the first and second waves of the pandemic.

## 2. Materials and Methods

### 2.1. Clinical Samples

Peripheral blood samples (*n* = 165) were obtained from unvaccinated COVID-19 patients at the Puerta del Mar University Hospital (*n* = 65), Puerto Real University Hospital (*n* = 10) and a nursing home associated with La Milagrosa Health Center in Jerez de la Frontera (*n* = 90) during the first and second waves of the pandemic (March to June and November to December of 2020, respectively) in Cadiz, Spain. Moreover, 40 samples obtained before 2020 (prepandemic samples) were retrieved from the Puerta del Mar University Hospital as negative controls. None of the research participants were vaccinated and no exclusion criteria were considered.

Peripheral blood samples were drawn by trained nursing staff from an intravenous cannula using an adaptor device (the BD Vacutainer Luer-Lok™ access device), to which vacuum tubes with a clot-activator collection tube and separating gel were attached. Samples were then promptly centrifuged at 2500× *g* for 10 min (at 23 °C) in a swinging-bucket centrifuge.

### 2.2. Bead Functionalization

Binding between SARS-CoV-2 S1 (ACROBiosystems SARS-CoV-2 (COVID-19) S1 protein, Mouse IgG2a Fc Tag), RBD (ACROBiosystems SARS-CoV-2 (COVID-19) S protein RBD, Mouse IgG2a Fc Tag) and NC (ACROBiosystems SARS-CoV-2 (COVID-19) nucleocapsid protein, His Tag) recombinant proteins to the carboxylated bead surface was based on the methodology described in the Bio-Plex Pro^TM^ magnetic COOH Beads Amine Coupling Kit (Bio-Rad, Hercules, CA, USA) instruction manual. Magnetic COOH beads (Bio-Rad, Hercules, CA, USA), subset #26, #36 and #34, were used to link the S1, RBD and NC, respectively. Beads were washed twice with PBS, and subsequently with Milli-Q water, then magnetically separated and resuspended in HBS-EP (Cytiva^®^). EDC (Pierce Biotechnology, Thermo Fisher Scientific, Waltham, MA, USA) and Sulfo-NHS (Pierce Biotechnology, Thermo Fisher Scientific, Waltham, MA, USA) reagents were then added and mixed for the activation and stabilization of the beads. Before carrying out two more washes with PBS (with a pH of 7.4), the activated beads were incubated at room temperature in the dark for 20 min. After incubation, the corresponding recombinant SARS-CoV-2 protein (S1 subunit, RBD or NC) was added at 5μM and incubated with stirring at room temperature for 2 h. After magnetic separation, the supernatant was removed and the beads were incubated with ethanolamine for 7 min to inactivate any unused NHS moiety; then, they were washed with PBS (with a pH of 7.4), followed by resuspension in a blocking buffer (PBS (with a pH of 7.4), 10%FBS, 0.02% azida) and stored at 4 °C until further use.

### 2.3. Multiplex Immunoassay

To perform the serological assays, we used the reagents included in the Bio-Plex Cytokine Reagent Kit (Bio-Rad Laboratories). Patient sera were placed on a 96-well plate in duplicate, then diluted 1:10 in a sample diluent buffer. Subsequently, a mixture containing the desired amount per well of the S1-functionalized beads (subset #26), RBD-functionalized beads (subset#36) and NC-functionalized beads (subset#34), diluted 1:10 in an assay buffer, was prepared and added to each well. The plate was incubated while stirring at 1100 rpm in the dark for 30 min at room temperature. After incubation, 3 washes with wash buffer were carried out. Biotinylated antihuman IgG antibody (Goat antihuman IgG (heavy chain), Antibodies-online) was added to each well and diluted 1:100 in a detection antibody buffer. Afterwards, the 96-well plate was incubated while stirring at 1100 rpm in the dark for 30 min at room temperature, followed by 3 washes with wash buffer.

Fluorescent streptavidin (streptavidin-PE, Bio-Rad Laboratories), diluted 1:100 in an assay buffer, was added to each well. Subsequently, the 96-well plate was incubated again while stirring at 1100 rpm in the dark for 30 min at room temperature, and 3 washes with wash buffer were subsequently carried out. Lastly, the plate was stirred for 30 s to dislodge the beads before detection by the Bio-Plex^TM^ array reader (Bio-Plex 200 Luminex System, Bio-Rad Laboratories) in the Core Biomedical Research Facility from Cadiz University, thus allowing the simultaneous identification of anti-S1, anti-RBD and anti-NC IgG in the patients’ samples (Figure 1).

### 2.4. Statistical Analysis

The ROC curve analysis was performed with GraphPad Prism (version 9.0). A total of 98 samples were used for the ROC analysis, of which 40 were prepandemic and 58 were pandemic samples from the Puerta del Mar University Hospital. Abbot Alinity SARS-CoV-2 CLIA immunoassays for anti-NC, anti-S IgG and IgM antibodies were used as reference serological assays [21], with a combined sensitivity and specificity of nearly 100%. As no reference data for anti-RBD antibodies were available at that time, anti-S IgG CLIA data were used as a reference. The availability of CLIA and multiplex results were used as inclusion criteria for the trial.

The Matplotlib package in Python 3 was used to elaborate the scatter plots in order to study the association between anti-S1, anti-NC and anti-RBD IgG values in COVID-19 patients with multiple clinical variables (sex, age, days since a positive PCR, severity of COVID-19 disease, comorbidities and body mass index (BMI)). This package was also used to evaluate the differences in the anti-S1, anti-NC and anti-RBD IgG values between prepandemic (*n* = 40) and COVID-19 patients (*n* = 165). The SciPy package in Python 3 was also used to carry out the Mann–Whitney U test, with a significance level of 5%. Inclusion criteria for the analyses were the availability of data for each patient at the time of data collection; therefore, the number of samples varies according to the variable studied.

## 3. Results

### 3.1. Multiplex Immunoassay Detects Anti-SARS-CoV-2 S1, RBD and NC IgG Antibodies with a High Specificity and Sensitivity

In order to determine the specificity and sensitivity of the multiplex immunoassay performed on the Bio-Plex^TM^ platform, as well as to establish the appropriate cut-off values for positivity, an ROC analysis was performed.

The empirical ROC curve analysis for the detection of anti-S1 IgG antibodies (Figure 2A) shows that the multiplex immunoassay performed is capable of detecting anti-S1 IgG antibodies with a high sensitivity (94.34%) and specificity (100%), with a calculated empirical ROC area under the curve (AUC) of 0.9849. The ROC curve analysis for the anti-RBD IgG antibody detection evidences that the immunoassay is able to detect the anti-RBD IgG antibodies with a 94.34% sensitivity and a 100% specificity (Figure 2B), with a calculated empirical ROC AUC of 0.9941. The empirical ROC curve analysis for the detection of the anti-NC IgG antibodies estimates that the multiplex assay is capable of detecting anti-NC IgG antibodies with a 97.96% sensitivity and a 91.84% specificity (Figure 2C), with a calculated empirical ROC AUC of 0.9696. Overall, the multiplex immunoassay performed through the Bio-Plex^TM^ technology displays a very high sensitivity (94.34–97.96%) and specificity (91.84–100%), with a 95% confidence interval.

Nonetheless, in order to determine the cut-off values for the positivity of the different antibodies, the Youden index was calculated. Therefore, it has been finally determined that the cut-off values for the positivity are 25,000 mean fluorescence intensities (MFIs) for the anti-S1 IgG antibodies, 20,000 MFIs for the anti-RBD IgG antibodies and 23,000 MFIs for the anti-NC IgG antibodies (some exceptions will be discussed later).

### 3.2. Multiplex Immunoassay Can Discriminate between Positive and Negative Samples

To analyze the ability of the multiplex immunoassay to correctly discriminate between positive and negative samples, SARS-CoV-2 anti-S1, anti-NC and anti-RBD IgG antibodies were assayed in the prepandemic samples (*n* = 40) and COVID-19 samples (*n* = 165). As shown in Figure 3, low MFI values below the determined cut-off for positivity were recorded in all prepandemic samples with a *p*-value < 0.001. Higher MFI values would correlate with high values of anti-S1, anti-NC or anti-RBD IgG antibodies, and low MFI values would indicate the absence, or lower values, of IgG antibodies against SARS-CoV-2. Although the MFI values for anti-S1 and anti-RBD IgG in prepandemic samples are distinctly negative, indicating no prior exposure to the specific antigens or any cross-reactivity, when analyzing the anti-NC IgG antibodies, positive MFI values are detected in some prepandemic samples, stressing the risk of relying on single-antigen detection.

### 3.3. Comparison between Anti-S1, Anti-RBD and Anti-NC IgG Antibody Values by Different Clinical Variables

After successfully establishing our antibody detection assay, we advanced to analyzing patient samples to glean insights into the antibody response. Our goal was to correlate the antibody detection results with a range of clinical parameters, including age groups, sex, days since a positive PCR, disease severity, body mass index (BMI) and comorbidities.

The number of samples in each analysis varied depending on the data available at the time of collection. Prepandemic samples were not used for these analyses.

#### 3.3.1. Antibody Responses by Age and Sex

When comparing the anti-S1, anti-NC and anti-RBD IgG antibody values by age groups (Figure 4) and sex (Figure 5), no statistically significant differences were found (*p*-value > 0.05).

#### 3.3.2. Antibody Responses by Days since a Positive PCR

When comparing the antibody values among the COVID-19 samples according to the days elapsed after a PCR detection, we found no significant differences for anti-RBD IgG (Figure 6A), while significant differences were found for anti-S1 IgG values (Figure 6B) between the samples drawn on days 0–30 and 91–180 after the PCR detection with those from days 181–343. For the anti-NC antibody (Figure 6C), significant differences were found among all day ranges, with the exception of values from samples obtained on days 0–30 vs. 31–90 since a positive PCR.

#### 3.3.3. Association between Increased SARS-CoV-2 IgG Antibody Response and Disease Severity

Analysis of the relationship between antibody values and COVID-19 severity showed significant differences (*p*-value < 0.05) for the anti-S1 (Figure 7B) and anti-NC (Figure 7C) antibody values between asymptomatic patients and those who presented with mild, moderate or severe illness. Likewise, we found significant differences among the anti-NC IgG values from asymptomatic patients and patients with critical illness. These results suggested that asymptomatic patients produce a decreased IgG humoral response compared to those experiencing symptoms.

The classification has been made according to the severity values of respiratory infections and their definitions (SARS-CoV-2) as extracted from the Technical Document “Clinical Management of COVID-19: Hospital Care” (Ministry of Health—Government of Spain).

#### 3.3.4. Negative Association between SARS-CoV-2 Antibody Values and Body Mass Index

Patients with a healthy weight produced a decreased anti-RBD (Figure 8A) and anti-S1 (Figure 8B) IgG humoral profile when compared to overweight and obese patients (*p* ≤ 0.05). However, no statistically significant differences were found in the anti-NC antibody values (Figure 8C).

#### 3.3.5. Antibody Responses According to Comorbidities

We additionally searched for a relationship between single comorbidities and the IgG antibody profile. Comorbidities in this study were high blood pressure (HBP), diabetes mellitus (DM), obesity, asthma and allergic rhinitis (AR). Although no statistically significant differences were found when comparing the anti-RBD (Figure 9A) and anti-S1 (Figure 9B) IgG antibody values, those who had AR exhibited a decreased anti-NC IgG humoral response against SARS-CoV-2 (Figure 9C).

## 4. Discussion

Serology tests provide information about the epidemiology of an infection, which make them a fundamental tool for COVID-19 spread control. These tests also provide insights into the protective immunity of individuals against COVID-19 [22,23,24], and can be used for SARS-CoV-2 vaccine assessment and the development of new therapeutic approaches [25,26,27,28,29,30].

In order to determine the sensitivity and specificity of the multiplex immunoassay designed in this study, an ROC curve analysis was carried out. All in all, the multiplex assay performed through the Bio-Plex^TM^ technology displayed a very high sensitivity (94.34–97.96%) and specificity (91.84–100%), with a 95% confidence interval.

Interestingly, the MFI values detected in the prepandemic samples were considerably higher for the anti-NC IgG than the anti-S1 and anti-RBD IgG (Figure 3), even exceeding the positivity cut-off values for some samples. It has already been shown that the viral nucleocapsid (N), that serves a general purpose and is not subjected to strong immunological pressure, is more conserved than the spike among coronaviruses, like those causing the common cold, to which there is a very strong exposure in the general population. A broad cross-reactivity against common cold coronaviruses has also been shown for T cell responses for the N protein [31,32,33]. This hypothesis is strongly supported by the study of Anderson et al. [17], since they found that antibodies against some prepandemic seasonal human coronaviruses cross-reacted with the SARS-CoV-2 spike, RBD and NC proteins (4.2%, 0.93% and 16.2% of the serum samples, respectively). They showed that, while these antibodies were boosted upon SARS-CoV-2 infection, these cross-reactive SARS-CoV-2 antibodies were not associated with immune protection. These findings further stress the rational that, in order to avoid false-positive diagnoses due to anti-NC IgG antibody cross-reactivity, it may be recommended to rely not only on the detection of anti-NC IgG, but also on the detection of IgG antibodies against multiple SARS-CoV-2 epitopes, such as those present on S1 or RBD.

Similar results have been reported from recent studies, in which multiplex immunoassays based on similar approaches were developed and SARS-CoV-2 cross reactive anti-NC IgG antibodies were also found. Interestingly, they reported that multiplex IgG antibody testing increased the specificity and sensitivity compared to single-antigen antibody detection [34,35,36].

We found no association between the IgG antibody values and age (Figure 4) or sex (Figure 5). Although several studies have analyzed the humoral response to SARS-CoV-2 according to these variables, results are not homogeneous and reach different conclusions [37,38,39,40].

Additionally, no substantial differences between anti-S1, anti-NC and anti-RBD IgG antibody values detected through the multiplex immunoassay were found depending on the number of days since the positive PCR test was performed (Figure 6). However, we noted that the anti-S1 IgG antibody values slightly decreased after 6 months following a positive PCR test, while the anti-NC IgG antibody values already decreased after 3 months. Several studies indicate that the concentration of the anti-SARS-CoV-2 IgG antibodies remains readily detectable and elevated for up to 3 months before gradually declining over time. However, in some cases, they can still be reliably detected up to 15 months, and may exhibit long-lasting persistence [41]. Another study reported that antibodies against SARS-CoV-2 sustained for at least 9 months [42].

When comparing the IgG values among patients with different disease severities, we found that symptomatic patients exhibited lower anti-RBD and anti-S1 IgG antibody values than those with mild, moderate or severe symptoms (Figure 7), including critical patients, when the anti-NC antibody values were studied. These results are in line with previous findings, where the correlation between the clinical profiles of individuals and the values of IgG antibodies specific to SARS-CoV-2 was found to be notably stronger in cases of severe illness [37,43,44,45,46,47].

In accordance with past studies [37,38,48], we detected increased anti-S1 and anti-RBD IgG antibody values in overweight or obese individuals when compared with those of normal weight (Figure 8). According to a recent report by the World Obesity Forum [49], there is a clear correlation between the prevalence of obesity in a country and the increased mortality rate associated with COVID-19, even after accounting for age and wealth factors. This suggests that, despite exhibiting strong humoral responses, individuals with higher BMI values are more susceptible to developing severe infectious diseases compared to those who have a healthy weight.

Finally, we studied whether there was an association between COVID-19 patients’ comorbidities and IgG antibody values (Figure 9). Although we did not find statistically significant differences for patients with high blood pressure, diabetes mellitus, obesity or asthma, we did find significant variations in anti-NC IgG antibody values for patients affected by allergic rhinitis (AR). The study of Xu et al. [50] reported that none of the included studies reported statistically significant differences between AR and non-AR patients, but their analysis indicated that allergic rhinitis was considered a comorbidity that was associated with reduced severity and lower rates of hospitalization among COVID-19 patients, which may be explained by our findings.

Nevertheless, since this study has a limited cohort size, future studies are needed to support these findings. Moreover, a larger analysis, including information about new SARS-CoV-2 variants of concern (VOC) and vaccinated immune responses, should be performed, as we only studied the IgG response profile of patients exposed to the circulating variants of the first and second waves in Spain, which mainly included the A.2, A.5, B.1 and B.1.177 lineages [51].

All in all, these findings evidence that the Bio-Plex multiplex immunoassay proposed herein is a robust and high-throughput SARS-CoV-2 immune response monitoring tool capable of simultaneously detecting anti-S1, anti-NC and anti-RBD IgG antibodies in serum with a very high sensitivity and specificity. Moreover, the multiplex SARS-CoV-2 immunoassay that we have described offers a versatile and flexible framework that allows for the inclusion of additional antigens specific to emerging SARS-CoV-2 variants, as well as for different applications, such as for tumor, inflammatory or infectious disease immunodiagnostics and surveillance.

## Figures and Tables

**Figure 1 microorganisms-11-02997-f001:**
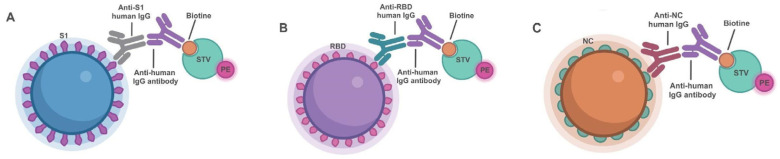
Schematic representation of the multiplex immunoassay developed for anti-S1, anti-RBD and anti-NC IgG antibody detection. Anti-S1 IgG (**A**), anti-RBD IgG (**B**) and/or anti-NC IgG (**C**) antibodies present in patients’ sera will bind to the magnetic beads functionalized with each antigen, and were differentially color-coded. Human antibodies captured onto each bead by the corresponding antigen would be recognized by the biotinylated antihuman IgG detection antibody, that will in turn bind fluorescently labeled streptavidin (STV-PE). PE fluorescence corresponding to the amount of antibodies captured by each antigen will be assigned by electronically gating each fluorescently coded bead, thus allowing for the simultaneous detection of IgG antibodies recognizing S1, NC and RBD by the Bio-Plex^TM^ array reader.

**Figure 2 microorganisms-11-02997-f002:**
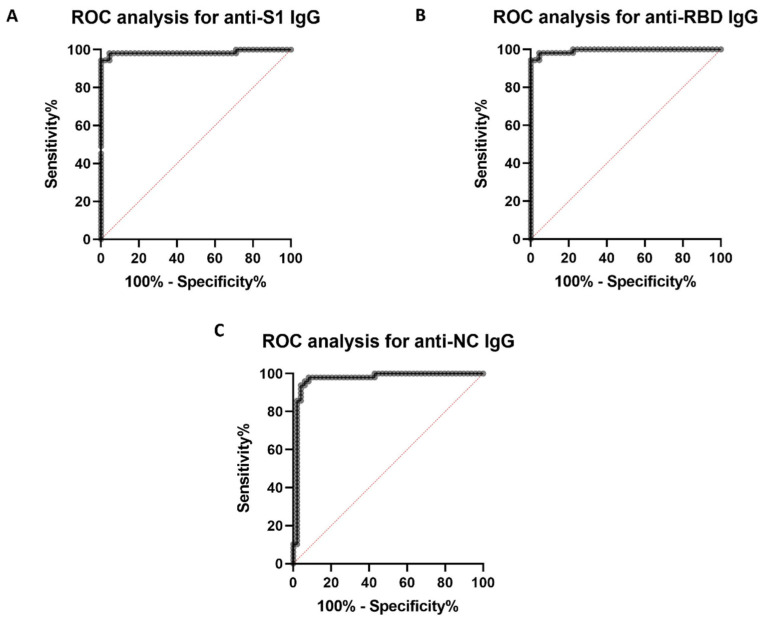
ROC analysis of the performed multiplex immunoassay. ROC curve representation (Black line) for the detection of anti-S1 IgG antibodies (**A**), anti-RBD IgG antibodies (**B**) and anti-NC IgG antibodies (**C**).

**Figure 3 microorganisms-11-02997-f003:**
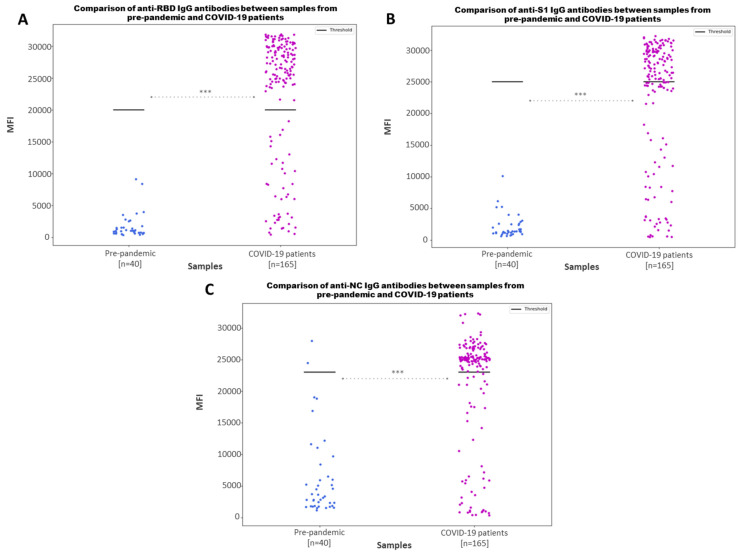
Comparison of antibody values between samples from prepandemic and COVID-19 patients. Scatter plot represents the MFI values for anti-RBD (**A**), anti-S1 (**B**) and anti-NC IgG (**C**). Bold lines represent the assay positivity cut-off value. The *p*-value returned from the Mann–Whitney U test carried out between both groups is shown as asterisks (***), for *p*-values ≤0.001.

**Figure 4 microorganisms-11-02997-f004:**
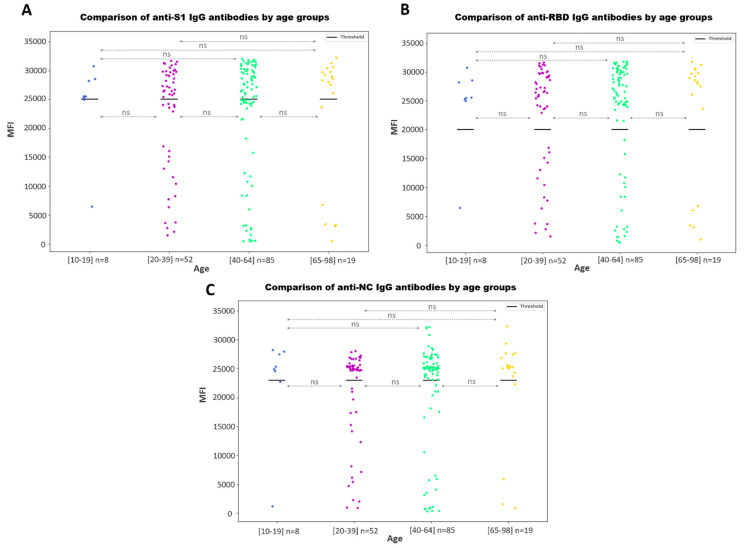
SARS-CoV-2 IgG antibody values by age groups. Scatter plot represents the MFI values for IgG anti-S1 (**A**), anti-RBD (**B**) and anti-NC IgG (**C**) antibodies. Bold lines represent the assay positivity cut-off. No statistically significant differences are represented with ‘ns’.

**Figure 5 microorganisms-11-02997-f005:**
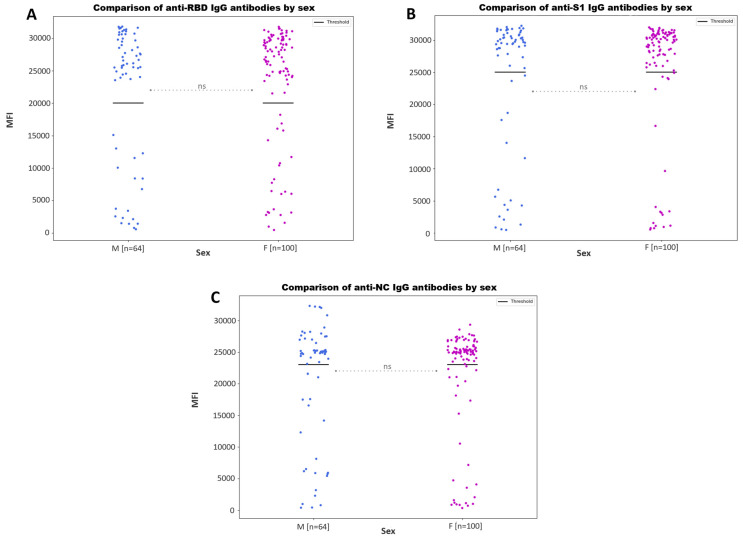
Scatter plot showing the IgG antibody values against SARS-CoV-2 obtained in the multiplex immunoassay according to sex. Scatter plot represents the MFI values for the detection of anti-S1 (**A**), anti-RBD (**B**) and anti-NC IgG (**C**) antibodies. Bold lines represent the assay positivity cut-off. No statistically significant differences are represented with ‘ns’.

**Figure 6 microorganisms-11-02997-f006:**
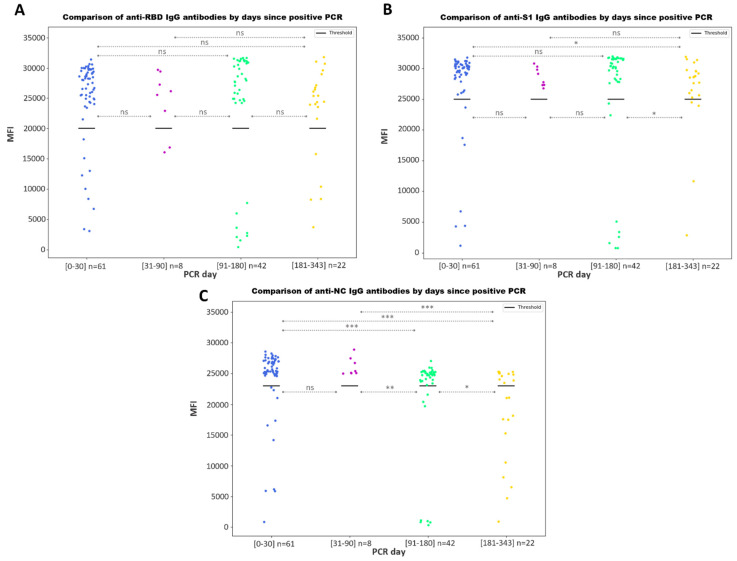
Dynamics of the SARS-CoV-2 antibody values according to the days since a positive PCR. Scatter plot representing the MFI values for the detection of anti-S1 (**A**), anti-RBD (**B**) and anti-NC IgG (**C**) antibodies. Bold lines represent the assay positivity cut-off. Statistical significance returned from the Mann–Whitney U test for each comparison is shown with asterisks (*), (**) or (***), for *p*-values ≤ 0.05, ≤0.01 and ≤0.001, respectively. No statistically significant differences are represented with ‘ns’.

**Figure 7 microorganisms-11-02997-f007:**
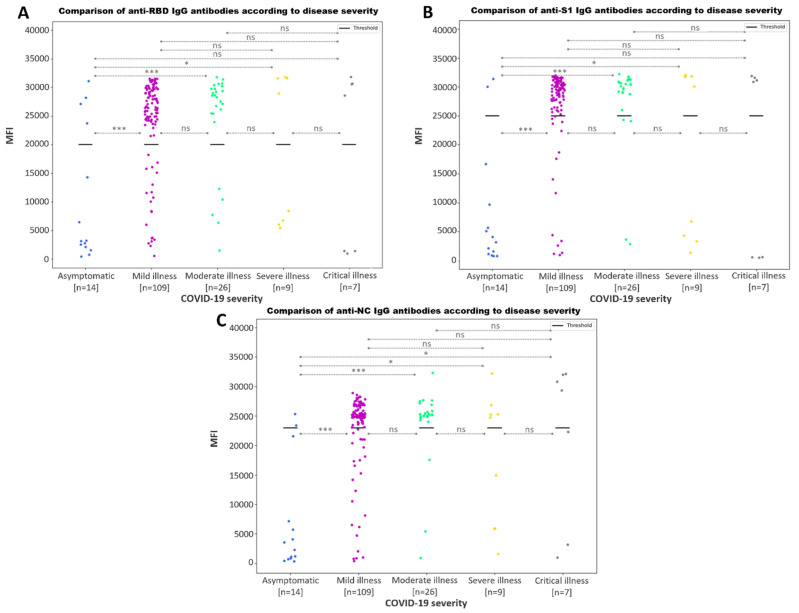
Comparison of the antibody values according to COVID-19 severity. Scatter plot represents the MFI values for anti-S1 (**A**), anti-RBD (**B**) and anti-NC IgG (**C**) antibodies. Bold lines represent the assay positivity cut-off. The *p*-values returned from the Mann–Whitney U test for each comparison is shown as asterisks (*) and (***), for *p*-values ≤ 0.05 and ≤0.001, respectively. No statistically significant differences are represented with ‘ns’.

**Figure 8 microorganisms-11-02997-f008:**
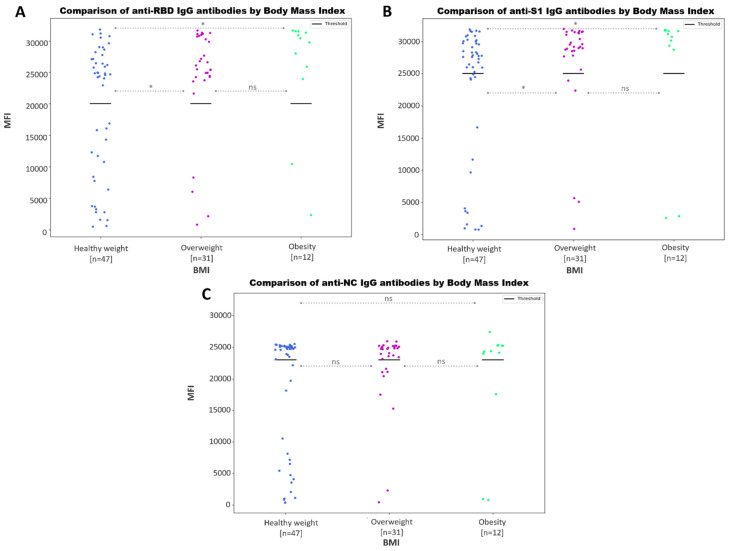
Comparison of different SARS-CoV-2 IgG antibody profiles by body mass index. Scatter plot represents the MFI values for anti-S1 (**A**), anti-RBD (**B**) and anti-NC IgG (**C**) antibodies. Bold lines represent the assay positivity cut-off. The *p*-values returned from the Mann–Whitney U test for each comparison is shown as asterisks (*), for *p*-values ≤ 0.05. No statistically significant differences are represented with ‘ns’.

**Figure 9 microorganisms-11-02997-f009:**
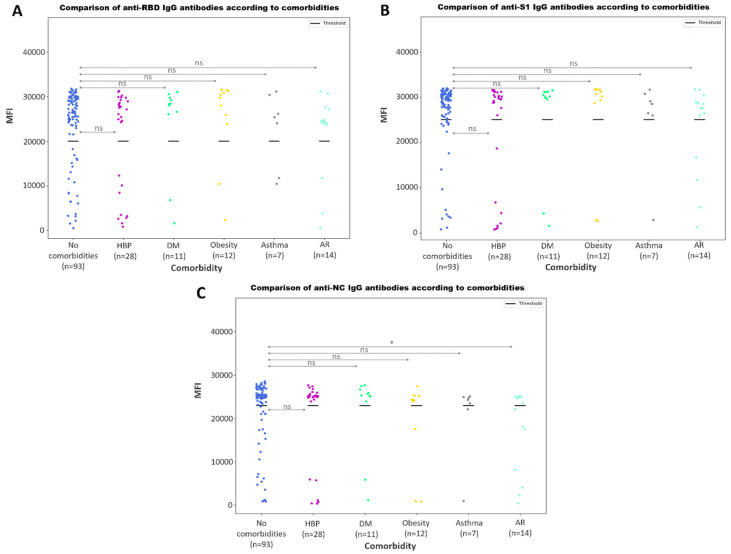
Comparison of antibody values according to comorbidities. Scatter plot represents the MFI values for anti-S1 (**A**), anti-RBD (**B**) and anti-NC IgG (**C**) antibodies. Bold lines represent the assay positivity cut-off. The *p*-values returned from the Mann–Whitney U test for each comparison is shown as asterisks (*), for *p*-values ≤ 0.05. No statistically significant differences are represented with ‘ns’.

## Data Availability

Data are contained within the article.

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
