# Peer review of "Development and Validation of a Highly Sensitive Multiplex Immunoassay for SARS-CoV-2 Humoral Response Monitorization: A Study of the Antibody Response in COVID-19 Patients with Different Clinical Profiles during the First and Second Waves in Cadiz, Spain"

_microorganisms, 2023, doi:10.3390/microorganisms11122997_

Round 1
Reviewer 1 Report
Comments and Suggestions for Authors
The paper entittled: Development and validation of a highly sensitive multiplex immunoassay for SARS-CoV-2 humoral response monitorization. Study of antibody response in COVID-19 patients with different clinical profiles during the first and second waves in Cadiz, Spain presents a comprehensive establishment of development of multiplex immunoassay for the detection of anti-S1, anti-NC and anti-RBD IgG antibodies. The study is properly designed and the manuscript clearly presents the results relevant in term of immunoassays application to detect and monitor specific antibodies against SARS-CoV-2. Authors comprehensively disscussed all relevant aspects of the study particullarly the need of further testing on VOCs distingueshed in the third and subsequent waves of SARS-Cov-2 infection. I reccomment to publish it in present form.
Author Response
Dear reviewer,
Thank you for your consideration of our manuscript. We sincerely appreciate your review and the time you have spent on it. Thank you for your comments.
Kind regards,
The authors.
Reviewer 2 Report
Comments and Suggestions for Authors
This study aimed to develop and validate a highly sensitive multiplex immunoassay for monitoring the humoral response to SARS-CoV-2, focusing on antibody responses in COVID-19 patients with diverse clinical profiles during the first and second waves of the pandemic in Cadiz, Spain.
Limitations
1. Limited Sample Diversity: The sample primarily consisted of unvaccinated individuals, which may not reflect the broader, vaccinated population.
2. Potential Cross-Reactivity: The potential for cross-reactivity in antibodies, especially anti-NC IgG, could impact the specificity of the assay.
3. Scope of Antigens: The study focused on specific antigens, potentially overlooking emerging variants.
4. Short Follow-up Period: The study's duration may not adequately capture long-term antibody persistence.
Detailed Suggestions
1. Expand Sample Population: Include both vaccinated and unvaccinated individuals from various demographics to enhance the study's applicability.
2. Enhanced Cross-Reactivity Analysis: Implement additional assays or computational methods to specifically identify and mitigate cross-reactivity issues.
3. Inclusion of Emerging Variants: Regularly update the assay to incorporate antigens from new SARS-CoV-2 variants to maintain its relevance.
4. Longer Follow-up Studies: Conduct longitudinal studies to track antibody levels over a more extended period, providing insights into the durability of the immune response.
5. Writing and Presentation: Improve the clarity and conciseness of the manuscript, ensuring that technical terms are well-explained for broader accessibility. Incorporate more visual aids, like charts or graphs, to effectively communicate complex data.
These improvements could significantly enhance the study's relevance, accuracy, and impact in the field of SARS-CoV-2 research.
Comments on the Quality of English LanguageImprove the clarity and conciseness of the manuscript, ensuring that technical terms are well-explained for broader accessibility.
Author Response
Dear reviewer,
Thank you for reviewing our manuscript. We appreciate your valuable comments and helpful suggestions. Below you will find the responses to each suggestion and the corresponding revisions highlighted in the submitted manuscript.
Reviewer's comments: Section “Limitations”
- Limited Sample Diversity: The sample primarily consisted of unvaccinated individuals, which may not reflect the broader, vaccinated population.
- Potential Cross-Reactivity: The potential for cross-reactivity in antibodies, especially anti-NC IgG, could impact the specificity of the assay.
- Scope of Antigens: The study focused on specific antigens, potentially overlooking emerging variants.
- Short Follow-up Period: The study's duration may not adequately capture long-term antibody persistence.
Authors' response to section “Limitations” (Items 1, 2, 3 and 4): We gratefully acknowledge the recommendations received. Actually, the manuscript focuses on the initial development of a multiplex assay and its validation, using a set of positive vs negative samples (pre-pandemic) for assay validation that on passing gives a glimpse on pre-vaccine populations. While it would be interesting to apply our test to current populations and long-term antibody monitoring, as well as to analyze emerging variants it falls out of the scope of the present manuscript as well as out of the protocol authorized by the Ethical Committee. Our assay is indeed perfectly amenable to addition of new variants coupled to differently coded beads (up to a hundred available) and such focus is actively pursued by collaborators.
Regarding cross-reactivity, we agree that anti-NC IgG detection shows the highest cross-reactivity but it is still at 91.84% specificity as shown in the result section. The overall test specificity is further enhanced by the concomitant determination of anti-RBD and anti-S antibodies where the assay has a 100% nominal specificity, highlighting the need for multiplex assays such as the one described here.
Reviewer's comments: Section “Detailed Suggestions”
Item 1. Expand Sample Population: Include both vaccinated and unvaccinated individuals from various demographics to enhance the study's applicability.
Authors' response to item 1 (Section “Detailed Suggestions”): The scope of the manuscript and sample recruitment approved by the Ethical committee in keeping with the project objectives was to develop a multiplex assay and to apply it to a limited number of samples to evaluate its performance.
We agree that the multiplex assay would be very useful to analyze and discern between vaccinated populations (positive only for Spike) and there is actually a collaborating group that has shown interest in the use of our assay.
Item 2. Enhanced Cross-Reactivity Analysis: Implement additional assays or computational methods to specifically identify and mitigate cross-reactivity issues.
Authors' response to item 2 (Section “Detailed Suggestions”): Cross-reactivity issues are mostly constrained to NC detection where specificity is down to 91.84% as opposed to 100% shown for Spike and RBD.
It has already been shown that the viral nucleocapsid (N), that serves a general purpose and is not subjected to strong immunological pressure, is more conserved than Spike among coronaviruses like those causing common cold to which there is a very strong exposure in the general population. A broad cross-reactivity against common cold coronaviruses has also been shown for T cell responses for the N protein [1–3]. We have added this paragraph to the discussion section.
References
- Bai, Z.; Cao, Y.; Liu, W.; Li, J. The SARS-CoV-2 Nucleocapsid Protein and Its Role in Viral Structure, Biological Functions, and a Potential Target for Drug or Vaccine Mitigation. Viruses 2021, 13, 1115, doi:10.3390/v13061115.
- Rak, A.; Donina, S.; Zabrodskaya, Y.; Rudenko, L.; Isakova-Sivak, I. Cross-Reactivity of SARS-CoV-2 Nucleocapsid-Binding Antibodies and Its Implication for COVID-19 Serology Tests. Viruses 2022, 14, 2041, doi:10.3390/v14092041.
- Westphal, T.; Mader, M.; Karsten, H.; Cords, L.; Knapp, M.; Schulte, S.; Hermanussen, L.; Peine, S.; Ditt, V.; Grifoni, A.; et al. Evidence for Broad Cross-Reactivity of the SARS-CoV-2 NSP12-Directed CD4+ T-Cell Response with Pre-Primed Responses Directed against Common Cold Coronaviruses. Front. Immunol. 2023, 14, 1182504, doi:10.3389/fimmu.2023.1182504.
Item 3. Inclusion of Emerging Variants: Regularly update the assay to incorporate antigens from new SARS-CoV-2 variants to maintain its relevance.
Authors' response to item 3 (Section “Detailed Suggestions”): We agree that regularly updating our assay, not only by replacing but by adding new variants coupled to differently coded beads (some 100 codes are available), would allow for a comprehensive study of the evolution of variant specific antibodies and indeed constitutes a strength of our assay, as the methods described here can be seamlessly applied to any new variant that emerges. Nonetheless, performing such studies are outside the scope of the present manuscript and would require a continuous sample recruitment that was not initially approved by the Ethical Committee.
Item 4. Longer Follow-up Studies: Conduct longitudinal studies to track antibody levels over a more extended period, providing insights into the durability of the immune response.
Authors' response to item 4 (Section “Detailed Suggestions”): We agree that the assay described in our manuscript may be used for tracking antibody levels over extended periods providing very valuable insights. In home developed assays such as the one described here are more affordable and thus allows for an extended use. Nonetheless, the purpose of the manuscript is to describe the development of the assay and to show its usefulness in an initial evaluation using a limited number of samples drawn before and after the immediate onset of the pandemics.
Item 5. Writing and Presentation: Improve the clarity and conciseness of the manuscript, ensuring that technical terms are well-explained for broader accessibility. Incorporate more visual aids, like charts or graphs, to effectively communicate complex data.
Authors' response to item 5 (Section “Detailed Suggestions”): We sincerely appreciate your suggestion and in order to follow up on it, we have rewritten the article not only improving the language but also making an effort to further clarify some issues ensuring broader availability.
Reviewer's comments: Section “Comments on the Quality of English Language”
Improve the clarity and conciseness of the manuscript, ensuring that technical terms are well-explained for broader accessibility.
Authors' response: We have rewritten the article not only improving the language but also making an effort to further clarify some issues ensuring broader availability.
Thank you for your consideration.
Kind regards,
The authors.
